# Sunitinib’s Effect on Bilateral Optic Nerve Damage in Rats Following the Unilateral Clamping and Unclamping of the Common Carotid Artery

**DOI:** 10.3390/biomedicines13030620

**Published:** 2025-03-03

**Authors:** Ibrahim Cicek, Cenap Mahmut Esenulku, Ahmet Mehmet Somuncu, Seval Bulut, Nurinisa Yucel, Tugba Bal Tastan, Taha Abdulkadir Coban, Halis Suleyman

**Affiliations:** 1Department of Ophtalmology, Faculty of Medicine, Erzincan Binali Yildirim University, Erzincan 24100, Turkey; ibrahim.cicek@erzincan.edu.tr; 2Department of Ophthalmology, Trabzon Kanuni Health Application and Research Center, Health Sciences University, Trabzon 61040, Turkey; cenapmahmut.esenulku@sbu.edu.tr (C.M.E.); mehmetsomuncu@hotmail.com (A.M.S.); 3Department of Pharmacology, Faculty of Medicine, Erzincan Binali Yildirim University, Erzincan 24100, Turkey; seval.bulut@erzincan.edu.tr; 4Pharmacy Services Program, Vocational School of Health Services, Erzincan Binali Yildirim University, Erzincan 24036, Turkey; nurinisa.yucel@erzincan.edu.tr; 5Department of Histology and Embryology, Faculty of Medicine, Erzincan Binali Yıldırım University, Erzincan 24100, Turkey; tbal@erzincan.edu.tr; 6Department of Biochemistry, Faculty of Medicine, Erzincan Binali Yıldırım University, Erzincan 24100, Turkey; acoban@erzincan.edu.tr

**Keywords:** sunitinib, optic nerve, ischemia–reperfusion, oxidative stress, inflammation

## Abstract

**Background/objectives**: Common carotid artery occlusion can cause oxidant and inflammatory damage to the optic nerve. In this study, the effect of sunitinib was investigated, the antioxidant and anti-inflammatory properties of which have been previously reported and shown to be protective in I/R injury and in preventing bilateral optic nerve ischemia–reperfusion (I/R) injuries after unilateral common carotid artery ligation in rats. **Methods**: In this study, 18 Albino Wistar male rats were divided into SG (sham-operated), CCU (clamping and unclamping), and SCCU (sunitinib + clamping and unclamping) groups. One hour before the surgical procedures, sunitinib (25 mg/kg, oral) was given to SCCU rats. Anesthesia was induced with ketamine (60 mg/kg, ip) and sevoflurane. The right common carotid arteries of all rats were accessed under anesthesia. While the skin opened in SG rats was closed with sutures, the right common carotid arteries of CCU and SCCU rats were clipped, and an ischemia period was created for 10 min. Then, reperfusion (6 h) was achieved by unclipping. After euthanasia with ketamine (120 mg/kg, intraperitoneally), the right and left optic nerves of the rats were removed and examined biochemically and histopathologically. **Results**: Malondialdehyde, tumor necrosis factor α, interleukin-1β, and interleukin-6 were increased, and total glutathione levels had decreased in both ipsilateral and contralateral optic nerves (*p* < 0.05). These changes were more prominent on the ipsilateral side. Similarly, histopathological damage was observed to be more on the ipsilateral side (*p* < 0.05). Biochemical and histopathological changes were significantly suppressed in rats receiving sunitinib treatment (*p* < 0.05). **Conclusions**: Sunitinib may protect optic nerve tissue against I/R injury by reducing oxidative stress and inflammation.

## 1. Introduction

Common carotid artery occlusion is usually associated with both internal and external carotid artery occlusions [1]. Embolism is known as one of the most common mechanisms of vascular occlusion [2]. A number of risk factors contribute to vascular occlusion, including hypertension, ischemic heart disease, dyslipidemia, diabetes mellitus, and smoking [1]. The existence of coronary artery disease, in particular, is closely associated with carotid artery stenosis in the context of systemic arteriosclerosis [3]. The complete occlusion of the internal carotid artery was reported to carry a high risk of ischemic complications in a previous study [4]. As blood supplies the ocular tissue from the internal carotid artery and, to a lesser degree, from external carotid artery branches [5], subtotal or total obstruction of the carotid artery may result in ocular ischemia [6]. Clinically common causes such as glaucoma and ischemic stroke may also lead to ischemia-induced optic nerve damage. Increased intraocular pressure leads to tissue ischemia, oxidative stress, inflammation, and, ultimately, glaucomatous neuropathy [7]. Reduced blood flow during ischemic stroke interferes with normal mitochondrial functioning and various triggered events, including reduced energy and oxidative stress, which can lead to the damage of retinal ganglion cells and the optic nerve [8]. Therefore, the first step in treating ischemic tissue should be to restore reperfusion [9]. The bilateral occlusion of the carotid artery induces oxidative stress in the retina of animals. Occlusion of the common carotid artery may also cause damage to the optic nerve [10]. According to another study, ligation of the common carotid artery results in an elevation in oxidative and proinflammatory cytokines as well as a decrease in antioxidant levels in the ocular tissue of rats [11]. In rats, an ischemia–reperfusion (I/R) damage model can be observed by occluding carotid arteries and subsequent reperfusion [12]. Carotid artery occlusion is induced by the ligation of the common carotid artery [11].

Sunitinib, the effects of which were evaluated on possible optic nerve injury in rats subjected to carotid artery ligation in this study, is a multikinase inhibitor drug that blocks some receptors such as the vascular endothelial growth factor receptor, platelet-derived growth factor receptor alpha and beta [13]. Sunitinib is approved for the treatment of renal cell carcinoma and imatinib-resistant gastrointestinal stromal tumors [14]. The impact of sunitinib on optic nerve injury resulting from common carotid artery occlusion has not been investigated. However, the neuroprotective effect of sunitinib has been previously investigated and shown to promote retinal ganglion cell survival in an experimental anterior ischemic optic neuropathy model [15]. A single subconjunctival injection of sunitinib–pamoate complex microcrystals was shown to provide neuroprotection in a rat optic nerve crush model [16]. The protective effect of sunitinib was also tested outside neuronal tissues, and it was shown to protect liver tissue from oxidative and inflammatory damage caused by I/R [17]. In addition, sunitinib was used in an experimental study with cisplatin, and it was reported that sunitinib alleviated nephrotoxicity associated with cisplatin-induced oxidative stress in addition to its tumor-inhibitory effect [18]. Therefore, this study aimed to evaluate whether sunitinib is effective in preventing bilateral optic nerve I/R injuries after unilateral common carotid artery ligation in rats both biochemically and histopathologically.

## 2. Materials and Methods

### 2.1. The Rats

In this study, 18 *Albino Wistar* male rats (290–305 g) were used. The rats were purchased from Erzincan Binali Yıldırım University Experimental Animals Application and Research Centre. For a week, the rats were housed in a laboratory with a temperature of 22 ± 1 °C and a 12 h periodic lighting system and fed ad libitum.

### 2.2. Drugs

Among the chemicals used in these experiments, sunitinib and ketamine were procured from Pfizer (Istanbul, Turkey). Sevoflurane was obtained from AbbVie (Istanbul, Turkey).

### 2.3. Experimental Animal Groups

The rats were separated into three groups (every group/6 rats): the sham-operated group (SG), the right common carotid clamping and unclamping operated group (CCU), and sunitinib + common carotid clamping and unclamping operated group (SCCU).

### 2.4. Surgical and Pharmacological Procedures

First, in the SCCU group, sunitinib at 25 mg/kg was administered orally one hour before anesthesia. The CCU and SG groups were given pure water orally at the same volume. Doses in this study were determined considering previous studies. Drug doses administered in experimental animals differed from human doses due to differences in body surface area and metabolism. Doses used in animal studies were converted to human doses by various methods [19]. Sunitinib was previously used at the same dose (25 mg/kg) in different experimental studies and showed antioxidant and anti-inflammatory activity [17,20,21]. Ketamine hydrochloride injections of 60 mg/kg were administered intraperitoneally (IP) for general anesthesia. During the surgical procedure, rats sniffed sevoflurane as needed to maintain anesthesia. An appropriate duration of anesthesia for surgery was considered to be when the animals were immobile in the supine posture [22]. Each rat was fixed to the operating table in the supine position during the operation. Incisions were made at the midline of the necks of the rats after the midline of the necks had been shaved and disinfected. In the following steps, a deep microdissection was performed following a superficial microdissection of the right common carotid artery. The right common carotid artery was located by visualizing the trachea and dissecting the paratracheal muscles, and a clip was placed on the artery. By keeping the clips closed for ten minutes, ischemia was induced. After this period, the clips were removed, the incisions were sutured, and reperfusion was administered for six hours [23]. Only a subcutaneous incision was performed on rats in the SG group. After this period, rats were sacrificed using ketamine (120 mg/kg, IP). Levels of oxidants, antioxidants, and proinflammatory cytokines such as malondialdehyde (MDA), total glutathione (tGSH), tumor necrosis factor α (TNF-α), interleukin-1β (IL-1β), and interleukin-6 (IL-6) were measured in the removed optic nerve tissue. Histopathological examinations were also conducted on the tissues.

### 2.5. Biochemical Analyses

#### 2.5.1. Specimen Preparation

The tissues were washed with saline. In a liquid nitrogen medium, the tissues were ground into powder and homogenized. The supernatants were separated and used to analyze MDA, tGSH, TNF-α, IL-1β, IL-6, and proteins.

#### 2.5.2. MDA, GSH, TNF-α, IL-1β and IL-6 and Protein Determination

Rat Enzyme-Linked ImmunoSorbent Assay (ELISA) kits obtained from Cayman Chemical Company were used for the determination of MDA (Cat no: 10009055) and GSH (Cat no: 703002) in optic nerve tissues. TNF-α (ng/mg protein), IL-1β (pg/mg protein), and IL-6 (ng/mg protein) were determined by ELISA kits obtained from Eastbiopharm Co., Ltd. (Hangzhou, China). Kit instructions were followed for each analysis. The tissue protein amount was tested using the Bradford method [24].

### 2.6. Histopathological Examination

For light microscopy evaluation, optic nerve tissue samples were fixed by adding 10% formaldehyde. After washing, they were treated with alcohol series for dehydration. They were made transparent with xylol and embedded in paraffin; around 4–5-micron sections were taken. These sections were stained with hematoxylin-eosin. After evaluation with Olympus DP2-SAL firmware software Ver.3.3.1.198 (Olympus^®^ Inc., Tokyo, Japan), photographs were taken. Histopathological changes in optic nerve tissue were defined as showing destruction, the presence of polymorphonuclear cells, increased thickness, increased astrocyte cell populations, and the presence of edema/vacuolization. Each sample was graded for each criterion as follows: 0, no damage; 1, mild damage; 2, moderate damage; and 3, severe damage. The pathologist performed the evaluation blind to the study groups.

### 2.7. Statistical Analysis

The statistical analyses were conducted using the IBM SPSS Statistical Program (IBM Corp., Version 22.0, Armonk, NY, USA). The figures were created using the GraphPad Prism program (Version 9.0.0, San Diego, CA, USA). One-way ANOVA was used to analyze independent biochemical findings. A post hoc Tukey test assessed intergroup differences. Since the right and left optic nerve biochemical findings of the same group were dependent data, a paired sample *t*-test was preferred for statistical analysis. Biochemical data were presented as the mean ± standard deviation (X ± SD). Sequential histopathological data were analyzed using the Kruskal–Wallis test. Dunn’s test was applied in the follow-up. Since the ordinal histopathological data of the right and left optic nerves of the same group were dependent data, the Wilcoxon signed-rank test was used for analysis. Histopathological data were presented as the median (quartile 1–quartile 3) and X ± SD. *p*-values less than 0.05 were interpreted significantly.

## 3. Results

### 3.1. Biochemical Results

#### 3.1.1. MDA Levels of Optic Nerve Tissues

As shown in Figure 1, MDA levels in the ipsilateral and contralateral optic nerve tissues of rats in the CCU were higher than those in the SG (*p* < 0.001). Sunitinib inhibited the carotid artery occlusion-induced increase in MDA in ipsilateral and contralateral optic nerves (*p* < 0.05). In addition, the MDA level in the ipsilateral optic nerves of CCU and SCCU was higher than that of the contralateral optic nerves (*p* < 0.05).

##### 3.1.2. tGSH Levels of Optic Nerve Tissues

As evident from Figure 2, the clamping and unclamping of the unilateral common carotid artery decreased the tGSH in the optic nerve tissue of the right and left optic nerves compared to the SG (*p* < 0.001). The tGSH levels in the right and left optic nerves in the SCCU were higher than in the CCU (*p* < 0.001). In the SCCU, the amount of tGSH in the left optic nerve was similar to SG (*p* = 0.820). The tGSH levels in the right optic nerve tissues of the CCU and SCCU were lower than those in the left (*p* < 0.05).

#### 3.1.3. TNF-α, IL-1β, and IL-6 Levels of Optic Nerve Tissues

The clamping and unclamping procedure applied to the unilateral common carotid artery increased TNF-α (Figure 3), IL-1β (Figure 4), and IL-6 (Figure 5) levels in the optic nerve tissue of both ipsilateral and contralateral optic nerves compared to the SG (*p* < 0.001). Sunitinib significantly inhibited a carotid artery occlusion-induced excessive increase in TNF-α, IL-1β, and IL-6 levels in the optic nerves of both ipsilateral and contralateral optic nerves (*p* < 0.05). TNF-α, IL-1β, and IL-6 levels in the left optic nerve and TNF-α and IL-6 levels in the right optic nerve in the SCCU group were similar to the side showing proinflammatory cytokine levels in the SG (*p* > 0.05). In terms of TNF-α, there was no difference between right and left TNF-α and IL-1β levels in the SCCU (*p* > 0.05), whereas in the CCU, the right optic nerve data were higher than the left ones (*p* < 0.05). IL-6 levels in the right optic nerve in the CCU and the left optic nerve in the SCCU were higher than the other optic nerves (*p* < 0.05).

### 3.2. Histopathologic Results

The outcomes gained from the statistical analysis of the histopathological findings of the right and left optic nerves are presented in Table 1. From the light microscopy examinations of ipsilateral optic nerve tissues from the SG, it was determined that the cells and the connective tissue surrounding the tissue were normal. Those areas consisting of axons were stained slightly eosinophilic, while locations containing glial cell nuclei were stained basophilic. A pathological appearance was not observed in astrocytes, their extensions, or the blood vessels located in nerve trabeculae (Figure 6A).

A significant amount of vacuolated tissue and edema were present in the ipsilateral optic nerve tissue sections of the CCU. There was a considerable increase in connective tissue surrounding the optic nerve. An infiltration of inflammatory cells was observed. A large number of hypertrophic and degenerating astrocytes were observed throughout the tissue. There were signs of congestion and dilatation in the blood vessels (Figure 6B).

According to the ipsilateral optic nerve tissue images obtained from the SCCU, the connective tissue around the tissue was similar to that seen in the healthy group. There was a significant decrease in vacuolization and edema in the tissue compared to the CCU. Astrocyte morphologies and blood vessels were similar to those of the sham-operated group (Figure 6C).

In the contralateral samples of the SG, no pathological findings were found in astrocytes and their extensions, blood vessels in the nerve trabeculae, or the connective tissue surrounding the optic nerve. Axons were stained eosinophilic, and glial cells were stained basophilic (Figure 7A).

In the CCU group’s contralateral optic nerve tissue sections, there were vacuolized areas and edema throughout the optic nerve. Hypertrophic and degenerated astrocytes, dilatation, and congestion in the blood vessels were evident (Figure 7B).

In general, the contralateral optic nerve tissue of the SCCU was similar to that of the healthy group. There was no evidence of vacuolized areas. The astrocytes and blood vessels appeared normal (Figure 7C).

In the optic nerve tissues of the CCU on the ipsilateral side, polymorphonuclear cell infiltration and the increase in the connective tissue thickness and edema were higher (*p* < 0.05) compared to the contralateral side, while destruction and the increase in astrocyte cell population levels were similar (*p* > 0.05).

In the ipsilateral optic nerves of the SCCU, connective tissue thickness, and edema were higher (*p* < 0.05) than the contralateral side, whereas the data in terms of tissue destruction and astrocyte cell population were similar (*p* > 0.05).

## 4. Discussion

In this study, oxidative stress and inflammatory responses were assessed in the optic nerve tissue of rats following unilateral occlusion of the common carotid artery and reperfusion. Furthermore, sunitinib was investigated biochemically and histopathologically for its protective effect against oxidative stress and inflammatory responses in the optic nerves. In the I/R model induced by the occlusion and reperfusion of the common carotid artery, an increase in oxidative stress and inflammatory markers was observed in both ipsilateral and contralateral optic nerve tissue, with greater levels on the ipsilateral side. Furthermore, our study showed that sunitinib pretreatment significantly prevented pathological changes in the right and left optic nerve tissues induced by the I/R procedure. The response of uninjured contralateral tissue to an ipsilateral lesion is called the mirror or contralateral effect. It has been reported that experimental perfusion reduction in one eye leads to decreased bilateral retinal function in healthy subjects [25]. It has been previously reported that severe unilateral internal carotid artery stenosis reduces orbital and ocular blood flow velocities on the ipsilateral and contralateral sides, with visual, functional, and morphological parameters similarly impaired in both eyes [26]. It is a known fact that hypoperfusion, followed by reperfusion, triggers oxidative stress and inflammation in related tissue [27]. It is an accepted view that unilateral lesions of the central nervous system or perfusion problems trigger an inflammatory response in the contralateral area. Contralateral retinas in models of unilateral retinal damage have been reported to show neuronal degeneration and glial activation [25]. Excessive glial activation threatens cell viability by creating an unstable proinflammatory environment that impairs neuronal survival. The activation of mediators of inflammation initiated in the damaged nerve and traveling from the optic chiasm to the contralateral optic nerve has been responsible for the contralateral response [25]. In this study, it can be said that the biochemical and histopathological findings of damage in the left optic nerve, in addition to the right optic nerve, are consistent with the literature.

It is known that ischemia is the deprivation of oxygen to tissues and organs as a result of decreased or lack of blood flow for various reasons [28]. It is after a critical period of ischemia that cell damage and/or death occurs, which varies based on the type of cell and the affected organ [29]. The process involves the production of high amounts of reactive oxygen species (ROS) from the mitochondria, endoplasmic reticulum, and Nicotinamide Adenine Dinucleotide Phosphate oxidase, as well as the sudden entry of oxygen into the cells [30]. These ROS attack cellular structures and initiate lipid peroxidation (LPO) and inflammation [31]. The excessive accumulation of ROS as a result of reperfusion results in an oxidative stress environment that disrupts oxidant–antioxidant homeostasis [32]. Among the most common and typical effects of oxidative stress is the development of LPO [33]. The MDA is an oxidant molecule that is produced as a secondary product during LPO and is highly toxic [34]. It is, therefore, used as a marker for oxidative stress and cell damage [35]. Our study found a significant increase in MDA levels in the ipsilateral and contralateral optic nerves of rats in the I/R group compared to the SG. It has recently been reported by Cicek et al. (2024) that LPO is induced in retinal tissue as I/R and MDA levels increase, causing oxidative damage [11]. The findings of our study are in agreement with those of other studies in the literature that have demonstrated tissue damage associated with increased MDA levels in the retina of rats following an I/R procedure [36,37,38]. The increase in MDA levels in both optic nerves was, however, inhibited significantly by sunitinib treatment in the animal groups. A review of the literature revealed that sunitinib reduced oxidative damage and inflammatory responses in a variety of tissue models, including the liver, the testis, and the ovary [17,20,21]. Our results support the hypothesis that oxidative stress in the optic nerve tissue after reperfusion causes tissue damage, and sunitinib has the potential to reduce this damage.

All living organisms have evolved a comprehensive set of antioxidant defenses to prevent the formation of free radicals or to limit their harmful effects [39]. The Müller (glial) cells of the retina contain a large amount of GSH, which may be beneficial to retinal neurons under circumstances of oxidative stress [40,41]. There is evidence in the literature that GSH plays an important role in the scavenging of ROS that are formed during I/R in a variety of tissues [42,43,44]. Hence, we measured the tGSH level in optic nerve tissue, which is an indicator of endogenous antioxidant mechanisms. It has been reported by Chen et al. (2015) that rats subjected to I/R have decreased GSH levels compared to a control group, which occurs simultaneously with an increase in oxidative stress [37]. Based on the findings of our study, we observed decreased tGSH levels in both optic nerves in the I/R group alone due to oxidative stress, which is consistent with the findings in the literature. Moreover, sunitinib treatment significantly suppressed this decrease in tGSH levels. Based on their study, Ince et al. (2021) demonstrated that sunitinib improves decreased antioxidant levels in ovarian tissue due to I/R-related oxidative damage [20]. Furthermore, some previous studies have indicated that sunitinib exhibits antioxidant activity by inhibiting a reduction in GSH levels in the liver [17] and testis [21] tissues after I/R. Based on these data, sunitinib may have the ability to protect optic nerve tissue from damage due to its antioxidant properties.

ROS accumulation during reperfusion causes inflammatory cells such as neutrophils to migrate to the region, and the activation of signaling pathways takes place as an infection response, and this process triggers additional cytokine and chemokine production [45,46]. Elevated levels of proinflammatory cytokines such as TNF-α, IL-1β, and IL-6 have been observed in various I/R studies [47,48,49]. In addition, it has been reported in the literature that increased proinflammatory cytokines accelerate the inflammatory process and negatively affect neuronal survival after retinal I/R [50]. TNF-α has important cellular signaling functions in many tissues and plays a role in systemic inflammation and apoptosis induction, and its levels increase in response to ischemic injury [11,51]. IL-1β is produced in conditions such as infection or tissue damage and triggers the production of various proinflammatory cytokines, including hypotension, fever, and IL-6 [52]. IL-6 is a multifunctional cytokine secreted by monocytes, B and T lymphocytes, fibroblasts, and activated macrophages during infection and tissue damage [53]. In light of this information, TNF-α, IL-1β, and IL-6 levels were shown in optic nerves in this study to investigate possible inflammatory processes after I/R. Our biochemical analyses indicated that there was a significant increase in TNF-α, IL-1β, and IL-6 levels in the optic nerve tissue of both ipsilateral and contralateral optic nerves in the I/R alone group. This increase was more noticeable, especially in the ipsilateral optic nerves. These findings indicate that after unilateral carotid artery ligation, the inflammatory response in the optic nerves increased, and cellular damage persisted. Furthermore, sunitinib treatment was observed to inhibit this excessive increase. In a recent study by Li et al. (2024), sunitinib was shown to alleviate immune cell infiltration by reducing the levels of TNF-α, IL-6, and IL-1β in hepatic I/R injury [54]. The anti-inflammatory activity of sunitinib has been reportedly demonstrated in various tissue models in the literature [17,20,21]. In another study, the neuroprotective effect of astaxanthin used against optic nerve ischemia was attributed to its antioxidant and anti-inflammatory activity. It was shown that astaxanthin decreased IL-1β and TNFα levels in the retina and protected the optic nerve tissue against oxidative stress and apoptosis [55]. In addition, sorafenib, a multikinase inhibitor, showed an anti-inflammatory effect by down-regulating the I/R-induced increase in proinflammatory mediators such as TNF-α in rat retinas [56]. The data obtained in this study indicate that sunitinib is a potential therapeutic agent in optic nerve I/R injury.

The histopathological findings in the optic nerve tissue are in agreement with the biochemical data. Changes in biochemical processes may cause histopathological damage by altering cellular functions. For example, ROS oxidize lipids in the cell membrane. They produce aldehydes such as MDA, which is itself toxic [11]. The attack on tissues is counteracted by antioxidant defense systems [9,34]. In this process, changes occur in antioxidant levels. In the case of the superiority of oxidant systems, an attack on macromolecules brings histopathological damage. As a result, the integrity of the cell membrane may be disrupted, histopathological tissue degeneration or inflammation may be observed, and cell death (necrosis or apoptosis) may develop [11]. In support of the literature, in this study, histopathological damage accompanied the increase in oxidant and proinflammatory cytokine levels and the decrease in antioxidants observed in the CCU group. In this study, it was observed that hypoperfusion followed by reperfusion caused tissue damage with marked vascular congestion, edema, astrocyte hypertrophy, vacuolization, and inflammatory cell infiltration. In the literature, it has been reported that conditions such as cerebral ischemia produce hypertrophic astrocytes and trigger reactive astrocytosis. Reactive astrocytes are also involved in the process by affecting ROS and various pro-/anti-inflammatory cytokine levels [57]. The optic nerve has been shown to be damaged following bilateral carotid ligation in previous studies [11]. There is documented evidence that this event significantly alters the structure of the neural retina and optic nerve [11,58]. Carotid artery ligation has also been reported to result in the degeneration of retinal tissue [11]. Many studies have demonstrated that inflammatory cell infiltration is associated with degeneration and signs of demyelination in optic nerve tissue in I/R [59]. Inflammatory cell infiltration is also known to play a role in the pathophysiology of reperfusion injury, especially through the mediators they secrete [Yapca]. Li et al. also pointed out that sunitinib decreased inflammatory cell activation and the release of inflammatory mediators [Li]. According to our study, the severity of histopathological damage was milder in the sunitinib group, whose oxidant and proinflammatory cytokine levels were comparable to those of the SG.

### Limitations

Dose-dependent studies, long-term evaluation, and functional outcome measures are important to assess the feasibility of sunitinib. Furthermore, the protective effect of sunitinib can be evaluated by specific cellular and molecular mechanisms. The limitations of this study include the lack of functional assessments such as visually evoked potentials or behavioral tests.

## 5. Conclusions

It is concluded that unilateral common carotid artery ligation significantly increases oxidants and proinflammatory cytokines in the contralateral optic nerves of rats and significantly decreases antioxidants. In ipsilateral optic nerves, oxidants and proinflammatory parameters increased, and antioxidant levels decreased more compared to the contralateral optic nerves. An analysis of the histopathological findings of the optic nerve tissue following I/R also confirmed the biochemical and structural indicators of damage. An inflammatory cell infiltration associated with cytokine production was observed in the ipsilateral optic nerve. According to our experimental results, unilateral carotid artery occlusion may affect the optic nerve tissue of both optic nerves to varying degrees. This study’s results suggest that sunitinib may be able to alleviate the effects of I/R injury by reducing oxidative stress and inflammation in optic nerve tissue. While sunitinib has potential as a treatment option to prevent optic neuropathy, further research is needed for clinical use.

## Figures and Tables

**Figure 1 biomedicines-13-00620-f001:**
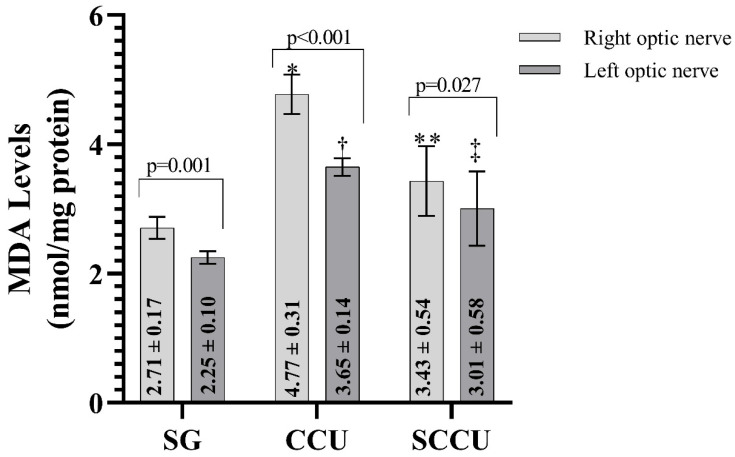
Right and left optic nerve MDA levels in the experimental groups. The bars show the mean ± standard deviation, n = 6. *, *p* < 0.001 vs. SG (right optic nerve); **, *p* < 0.001 vs. CCU (right optic nerve); ^†^, *p* < 0.001 vs. SG (left optic nerve); ^‡^, *p* < 0.05 vs. CCU (left optic nerve). MDA, malondialdehyde; SG, sham-operated group; CCU, right common carotid clamping and unclamping operated group; SCCU, sunitinib + common carotid clamping and unclamping operated group.

**Figure 2 biomedicines-13-00620-f002:**
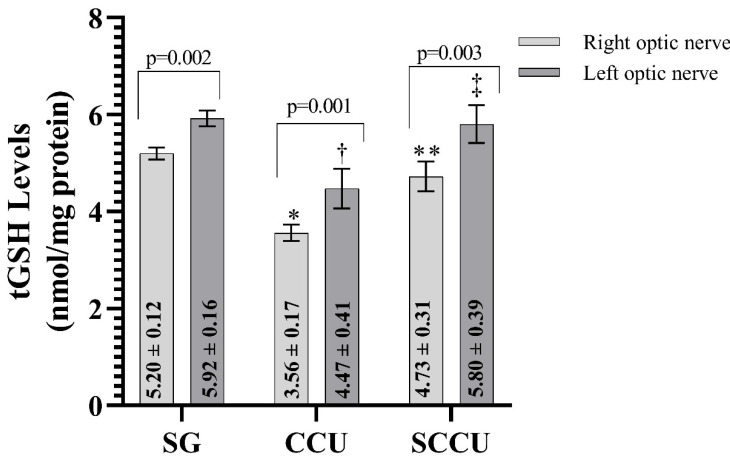
Right and left optic nerve tGSH levels in experimental groups. Bars show mean ± standard deviation, n = 6. *, *p* < 0.001 vs. SG (right optic nerve); **, *p* < 0.001 vs. CCU (right optic nerve); ^†^, *p* < 0.001 vs. SG (left optic nerve); ^‡^, *p* < 0.001 vs. CCU (left optic nerve). tGSH, total glutathione; SG, sham-operated group; CCU, right common carotid clamping and unclamping operated group; SCCU, sunitinib + common carotid clamping and unclamping operated group.

**Figure 3 biomedicines-13-00620-f003:**
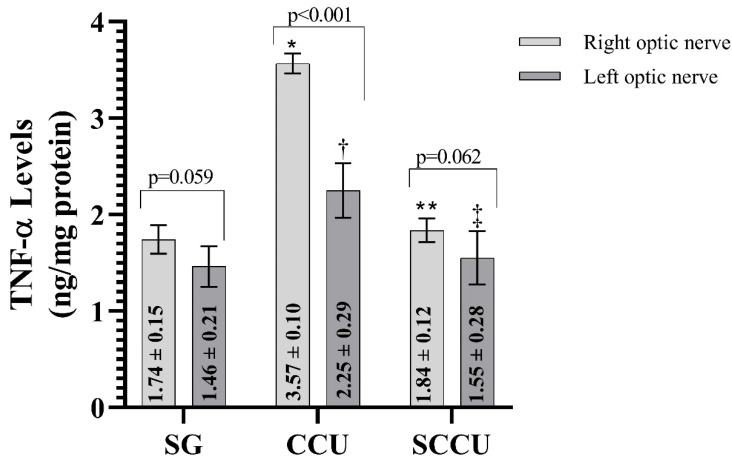
Right and left optic nerve TNF-α levels in experimental groups. Bars show mean ± standard deviation, n = 6. *, *p* < 0.001 vs. SG (right optic nerve); **, *p* < 0.001 vs. CCU (right optic nerve); ^†^, *p* < 0.001 vs. SG (left optic nerve); ^‡^, *p* < 0.001 vs. CCU (left optic nerve). TNF-α, tumor necrosis factor α; SG, sham-operated group; CCU, right common carotid clamping and unclamping operated group; SCCU, sunitinib + common carotid clamping and unclamping operated group.

**Figure 4 biomedicines-13-00620-f004:**
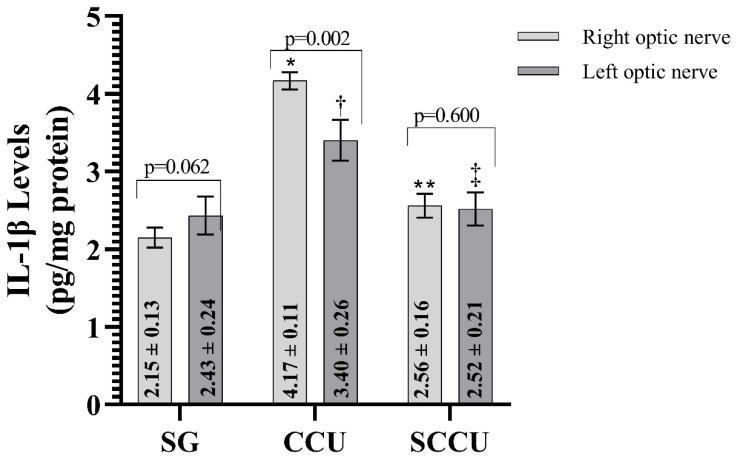
Right and left optic nerve IL-1β levels in experimental groups. Bars show mean ± standard deviation, n = 6. *, *p* < 0.001 vs. SG (right optic nerve); **, *p* < 0.001 vs. CCU (right optic nerve); ^†^, *p* < 0.001 vs. SG (left optic nerve); ^‡^, *p* < 0.001 vs. CCU (left optic nerve). IL-1β, interleukin-1β; SG, sham-operated group; CCU, right common carotid clamping and unclamping operated group; SCCU, sunitinib + common carotid clamping and unclamping operated group.

**Figure 5 biomedicines-13-00620-f005:**
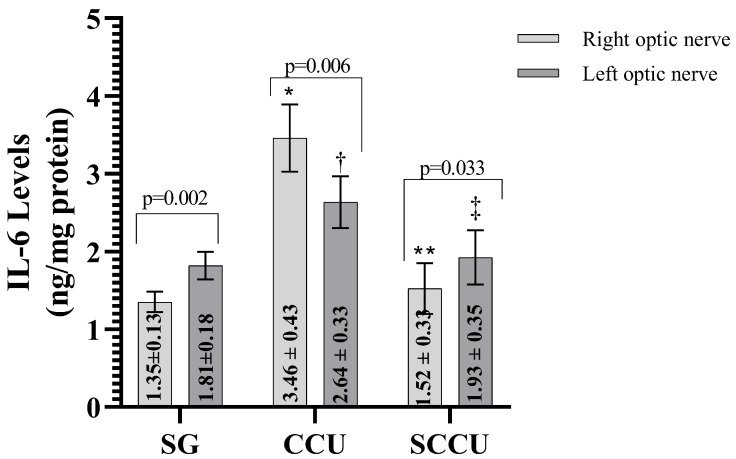
Right and left optic nerve IL-6 levels in experimental groups. Bars show mean ± standard deviation, n = 6. *, *p* < 0.001 vs. SG (right optic nerve); **, *p* < 0.001 vs. CCU (right optic nerve); ^†^, *p* < 0.05 vs. SG (left optic nerve); ^‡^, *p* < 0.05 vs. CCU (Left optic nerve). IL-6; interleukin-6; SG, sham-operated group; CCU, right common carotid clamping and unclamping operated group; SCCU, sunitinib + common carotid clamping and unclamping operated group.

**Figure 6 biomedicines-13-00620-f006:**
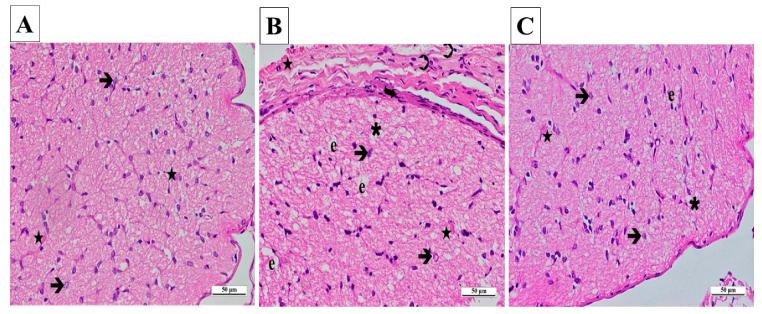
(**A**–**C**) Histopathological appearances of the right optic nerves of the experimental groups (H&E ×400). (**A**) SG group: ➔, astrocyte; ★, capillary vessel view. (**B**) CCU group: ➔, hypertrophied and degenerated astrocyte; ★, vascular dilatation and congestion; **∗**, vacuolization; ➲, inflammatory cell; e, edema; curved arrow, increase in thickness. (**C**) SCCU group: →, hypertrophic and degenerated astrocyte; ★, mild vascular congestion and dilatation; **∗**, vacuolization; e, edema. SG, sham-operated group; CCU, right common carotid clamping and unclamping operated group; SCCU, sunitinib + common carotid clamping and unclamping operated group.

**Figure 7 biomedicines-13-00620-f007:**
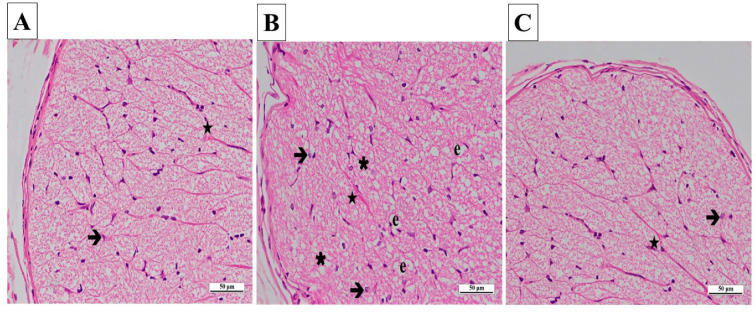
(**A**–**C**) Histopathological appearances of the left optic nerves of the experimental groups (H&E ×400). (**A**) SG group: ➔, astrocyte; ★, blood capillary, ×400. (**B**) CCU group: ➔, hypertrophied and degenerated astrocyte; ★, vascular congestion and dilatation; **∗**, vacuolization; e, edema. (**C**) SCCU group: →, astrocyte; ★, blood capillary. SG, sham-operated group; CCU, right common carotid clamping and unclamping operated group; SCCU, sunitinib + common carotid clamping and unclamping operated group.

**Table 1 biomedicines-13-00620-t001:** Grading data of histopathological damage in the experimental groups.

Histopathological Parameters	Optic Nerves	SG	CCU	SCCU
		Median (Quartile 1–Quartile 3) X ± SD		
Destruction	Right	0 (0–0) 0 ± 0	3 (2–3) * 2.5 ± 0.56	0 (0–1) ** 0.42 ± 0.55
	Left	0 (0–0) 0 ± 0	3 (2–3) ^†^ 2.53 ± 0.56	0 (0–1) ^‡^ 0.42 ± 0.55
Polymorphonuclear Cell Infiltration	Right	0 (0–0) 0 ± 0	1 (0–1) * 0.83 ± 0.70	0 (0–0) ** 0 ± 0
	Left	0 (0–0) 0 ± 0	0 (0–0) ^†^ 0.11 ± 0.32	0 (0–0) ^‡^ 0 ± 0
Increase In Connective Tissue Thickness	Right	0 (0–0) 0 ± 0	2 (2–3) * 2.28 ± 0.66	0 (0–1) ** 0.44 ± 0.56
	Left	0 (0–0) 0 ± 0	1 (1–1) ^†^ 0.94 ± 0.63	0 (0–0) ^‡^ 0.17 ± 0.38
Increase In Astrocyte Cell Population	Right	0 (0–0) 0 ± 0	2 (1–2) * 1.61 ± 0.60	0 (0–1) ** 0.42 ± 0.50
	Left	0 (0–0) 0 ± 0	2 (1–2) ^†^ 1.67 ± 0.68	0 (0–1) ^‡^ 0.42 ± 0.50
Edema/Vacuolization	Right	0 (0–0) 0 ± 0	3 (2–3) * 2.58 ± 0.50	0 (0–1) ** 0.42 ± 0.50
	Left	0 (0–0) 0 ± 0	3 (2–3) ^†^ 2.31 ± 0.62	0 (0–0.75) ^‡^ 0.25 ± 0.44

Histopathological grading: 0, no damage; 1, mild damage; 2, moderate damage; 3, severe damage. *, *p* < 0.05 vs. SG (right optic nerve); **, *p* < 0.05 vs. CCU (right optic nerve); ^†^, *p* < 0.05 vs. SG (left optic nerve); ^‡^, *p* < 0.05 vs. CCU (left optic nerve). SG, sham-operated group; CCU, right common carotid clamping and unclamping operated group; SCCU, sunitinib + common carotid clamping and unclamping operated group.

## Data Availability

The original contributions presented in this study are included in the article. Further inquiries can be directed to the corresponding authors.

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
