# Peer review of "Sunitinib’s Effect on Bilateral Optic Nerve Damage in Rats Following the Unilateral Clamping and Unclamping of the Common Carotid Artery"

_biomedicines, 2025, doi:10.3390/biomedicines13030620_

Round 1
Reviewer 1 Report
Comments and Suggestions for Authors
This paper investigated the effect of sunitinib on bilateral optic nerve ischemia-reperfusion (I/R) injury in rats caused by unilateral common carotid artery ligation and found that sunitinib may protect optic nerve tissue by reducing oxidative stress and inflammation. This study has certain clinical value. However, there are still the following problems in this article:
1. If only a single dose of sunitinib is used in the study, does it take into account the different effects of different doses of sunitinib on the experimental results? Generally, high, medium, and low doses are required.
2. What is the basis for the choice of sunitinib dosage?
3. Only the relevant indicators of oxidative stress and inflammatory response were studied, and the specific cellular and molecular mechanisms of sunitinib's protective effect were not deeply explored.
4. Sunitinib is a class of anti-tumor drugs that can selectively target a variety of receptor tyrosine kinases. How to understand that sunitinib both inhibits tumor cells and protects normal cells?
5. Why are there significant differences in the left and right optic nerves between the sham group?
6. A short-term reperfusion regimen was used, and only 6 hours after reperfusion was observed. Is the effect of reperfusion on the long term considered to be considered?
7. It is suggested that the discussion could be more in-depth.
Author Response
Reviewer 1
- If only a single dose of sunitinib is used in the study, does it take into account the different effects of different doses of sunitinib on the experimental results? Generally, high, medium, and low doses are required.
Response: You are right about this. Sunitinib was previously used by our team in different studies and showed antioxidant and anti-inflammatory activity at the same dose (1-3). Therefore, a single-dose study was preferred.
1-Bilici, S.; Yazici, G.N.; Altuner, D.; Aggul, A.G, Suleyman H. Effect of Sunitinib on Liver Oxidative and Proinflammatory Damage Induced by Ischemia-Reperfusion in Rats. Transplant Proc. 2021, 53, 2140-2146.
2-Keskin, E.; Erdogan, A.; Suleyman, H.; Yazici, G.N..; Sunar, M.; Gul, M.A. Effect of sunitinib on testicular oxidative and proinflammatory damage induced by ischemia-reperfusion in rats. Rev. Int. Androl. 2022, 20 Suppl 1, S17-S23.
2-İnce, S.; Özer, M.; Göktuğ Kadıoğlu, B.; Kuzucu, M.; Gündoğdu, B.; Gürsul, C.; Süleyman Z, Suleyman, H. Biochemical and Histopathological Evaluation of Sunitinib Effect on Ovarian Injuries by Ischemia-Reperfusion in Rats. Int. J. Pharmacol. 2021, 17, 65-72.
- What is the basis for the choice of sunitinib dosage?
Response: Dosage determination was performed based on literature data. Sunitinib has been previously used by our team in different studies and has shown antioxidant and anti-inflammatory activity at the same dose. Additional references regarding the text and dose have been added to the experimental procedure section. Below sentence added to the text for sunitinib dose.
‘Sunitinib was previously used at the same dose in different studies and showed antioxidant and anti-inflammatory activity.’
1-Bilici, S.; Yazici, G.N.; Altuner, D.; Aggul, A.G, Suleyman H. Effect of Sunitinib on Liver Oxidative and Proinflammatory Damage Induced by Ischemia-Reperfusion in Rats. Transplant Proc. 2021, 53, 2140-2146.
2-Keskin, E.; Erdogan, A.; Suleyman, H.; Yazici, G.N..; Sunar, M.; Gul, M.A. Effect of sunitinib on testicular oxidative and proinflammatory damage induced by ischemia-reperfusion in rats. Rev. Int. Androl. 2022, 20 Suppl 1, S17-S23.
3-İnce, S.; Özer, M.; Göktuğ Kadıoğlu, B.; Kuzucu, M.; Gündoğdu, B.; Gürsul, C.; Süleyman Z, Suleyman, H. Biochemical and Histopathological Evaluation of Sunitinib Effect on Ovarian Injuries by Ischemia-Reperfusion in Rats. Int. J. Pharmacol. 2021, 17, 65-72.
- 3. Only the relevant indicators of oxidative stress and inflammatory response were studied, and the specific cellular and molecular mechanisms of sunitinib's protective effect were not deeply explored.
Response: Yes, you are right. It is certainly important to examine the protective effect of sunitinib at the level of specific cellular and molecular mechanisms. However, the fact that our study did not have a funder and all expenses were covered by the authors was a limiting factor for us.
- Sunitinib is a class of anti-tumor drugs that can selectively target a variety of receptor tyrosine kinases. How to understand that sunitinib both inhibits tumor cells and protects normal cells?
Response: The point you mentioned is quite important from a clinical perspective. The design of our study was not designed in a way that we could understand that sunitinib both inhibits tumor cells and protects normal cells. We did a literature review. Suddek's study drew attention to the antitumor and normal tissue protective effects of sunitinib. Therefore, we thought of adding the data from this study in the introduction section and added the following sentence to the introduction section.
‘In addition, sunitinib was used in an experimental study with cisplatin and it was reported that sunitinib alleviated nephrotoxicity associated with cisplatin-induced oxidative stress in addition to its tumor inhibitory effect.’
Suddek G. M. (2011). Sunitinib improves chemotherapeutic efficacy and ameliorates cisplatin-induced nephrotoxicity in experimental animals. Cancer chemotherapy and pharmacology, 67(5), 1035–1044. https://doi.org/10.1007/s00280-010-1402-1
- Why are there significant differences in the left and right optic nerves between the sham group?
Response: Although the data of the sham group were close to each other, differences were found in some data. In our study, MDA was lower, tGSH and IL-6 were higher, TNF-α and IL-1β values were similar in the contralateral side of the sham group compared to the ipsilateral side. In fact, inflammatory parameters should have been higher on the ipsilateral side where oxidative stress data were more prominent. We performed a literature review on the reasons for these differences. However, we could not find any study that could provide an explanation. We also thought that it might be related to the sensitivity of the statistical test. The Paired Sample t-Test we chose was a test used in cases with 2 dependent data.
- A short-term reperfusion regimen was used, and only 6 hours after reperfusion was observed. Is the effect of reperfusion on the long term considered to be considered?
Response: We used literature for the study design. But, you are right that long-term effects after reperfusion could also be evaluated. Thank you for this reminder. We will definitely take this into account in our future studies.
Emir, I., Suleyman, Z., & Suleyman, H. (2024). Effect of thiamine pyrophosphate on oxidative damage in the brain and heart of rats with experimentally induced occlusion of the common carotid artery. Investigación Clínica, 65(2), 220-229].
Musayeva, A., Unkrig, J. C., Zhutdieva, M. B., Manicam, C., Ruan, Y., Laspas, P., Chronopoulos, P., Göbel, M. L., Pfeiffer, N., Brochhausen, C., Daiber, A., Oelze, M., Li, H., Xia, N., & Gericke, A. (2021). Betulinic Acid Protects from Ischemia-Reperfusion Injury in the Mouse Retina. Cells, 10(9), 2440. https://doi.org/10.3390/cells10092440
- It is suggested that the discussion could be more in-depth.
Response: Additions were made to the discussion section.

Reviewer 2 Report
Comments and Suggestions for Authors
The manuscript by Ibrahim Cicek and co-workers describes investigation of the protective effects of sunitinib on optic nerve injury following carotid artery occlusion. This research question is highly relevant and fills an important gap in current knowledge regarding neuroprotective strategies. The experimental work was carefully planned with appropriate control groups (sham, CCU, and SCCU) and the methodology combining both biochemical and histopathological analyses to provide sufficient evidence for the findings. The authors demonstrated bilateral effects of unilateral carotid occlusion and clearly show the protective effects of sunitinib. The results are important and deserve publication.
I have just the following technical comments:
Figure 5: The dot is missing in the standard deviation shown on one of the bars.
Figure 6: Arrows, asterisks, and inscriptions are hardly visible in the pictures. Also, some characters near the semicolons are omitted in the figure caption. The same applies to Figure 7.
I recommend acceptance of the manuscript for publication after minor revision.
Author Response
Reviewer 2
Figure 5: The dot is missing in the standard deviation shown on one of the bars.
Response: We thank you for your attention review. The mistake was corrected.
Figure 6: Arrows, asterisks, and inscriptions are hardly visible in the pictures. Also, some characters near the semicolons are omitted in the figure caption. The same applies to Figure 7.Resimlerde oklar, yıldızlar ve yazılar neredeyse hiç görünmüyor.
Response: Figure 6 and 7 are revised.

Reviewer 3 Report
Comments and Suggestions for Authors
several areas require revision to improve clarity, scientific rigor, and overall presentation.
Major Comments
I suggest to include additional background on the broader clinical relevance of optic nerve ischemia , example: its relation to glaucoma or stroke-induced optic neuropathy).
Discuss the recent advances in the use of multikinase inhibitors like sunitinib for neuroprotection in ophthalmology
It is important to include referencing studies on sunitinib’s effects on retinal or central nervous system ischemia models for better contextualization.
Methodology
Justify the choice of sunitinib dose (25 mg/kg)
Were different doses tested in pilot studies? How does this dose compare with clinical equivalents?
The reperfusion period of 6 hours is relatively short for assessing long-term neuroprotection. Would longer observation periods provide more translationally relevant insights?
The study lacks functional assessments such as visual evoked potentials or behavioral tests that could have strengthened the impact of findings beyond biochemical and histopathological analysis.
Results Interpretation
The biochemical findings are clear, but the statistical power of comparisons should be explicitly stated. Were power calculations performed for the sample size?
The study demonstrates increased contralateral optic nerve damage, which is a crucial finding. The authors should expand on the mechanism of mirror-image injury in I/R models, possibly citing recent work on cross-neuronal inflammatory signaling.
The histopathological findings should be correlated better with biochemical markers. For instance, how does astrocyte proliferation relate to oxidative stress markers like MDA?
Discussion
While the discussion is well-structured, it would benefit from a comparative analysis with other neuroprotective agents tested in optic nerve ischemia (e.g., anti-inflammatory drugs, antioxidants, or VEGF inhibitors).
The limitations section should address the need for dose-dependent studies, long-term evaluation, and functional outcome measures to assess the real-world applicability of sunitinib.
Minor Comments:
Line 22-23: The study objective should be more precise, emphasizing sunitinib’s specific mechanisms of action.
Line 81-82: Manufacturer details of sevoflurane should be double-checked for accuracy.
Table 1: Ensure consistent formatting in the presentation of statistical significance markers (p-values).
References: Some references are outdated (e.g., studies before 2010). Consider incorporating recent publications (2020-2024) on optic nerve I/R injury and pharmacological interventions.
Author Response
Reviewer 3
Major Comments
-I suggest to include additional background on the broader clinical relevance of optic nerve ischemia , example: its relation to glaucoma or stroke-induced optic neuropathy).
Response: The following text has been added to the introduction based on your suggestion. Clinically common causes such as glaucoma and ischaemic stroke may also lead to ischemia-induced optic nerve damage. Increased intraocular pressure leads to tissue ischemia, oxidative stress, inflammation, and ultimately glaucomatous neuropathy [Coviltir, V., Burcel, M. G., Baltă, G., & Marinescu, M. C. (2024). Interplay Between Ocular Ischemia and Glaucoma: An Update. International Journal of Molecular Sciences, 25(22), 12400]. Reduced blood flow during ischaemic stroke interferes with normal mitochondrial functioning and various triggered events, including reduced energy and oxidative stress, can lead to damage of retinal ganglion cells and the optic nerve [Kingsbury, C., Heyck, M., Bonsack, B., Lee, J. Y., & Borlongan, C. V. (2020). Stroke gets in your eyes: stroke-induced retinal ischemia and the potential of stem cell therapy. Neural Regeneration Research, 15(6), 1014-1018.].
-Discuss the recent advances in the use of multikinase inhibitors like sunitinib for neuroprotection in ophthalmology.
Response: Additions were made to the discussion section.
-It is important to include referencing studies on sunitinib’s effects on retinal or central nervous system ischemia models for better contextualization.
Response: Studies on the neuroprotective effect of sunitibib against ischemia were added to the introduction section. The following text has been added to the introduction section.
However, the neuroprotective effect of sunitinib has been previously investigated and shown to promote retinal ganglion cell survival in an experimental anterior ischemic optic neuropathy model [Sharma, S.M.; Wu-chen, W.Y.; Yang, Z.; Vilson, F.L.; Guo, Y.; Miller, N.R.; Don-ald, Z.J.; Bernstein, S.L. (2011). Sunitinib Malate Preserves Retinal Ganglion Cells in Rodent NAION. Invest. Ophthalmol. Vis. Sci. 2011, 52, 6614-6614]. A single subconjunctival injection of sunitinib-pamoate complex microcrystals was shown to provide neuroprotection in a rat optic nerve crush model [Hsueh, H. T., Kim, Y. C., Pitha, I., Shin, M. D., Berlinicke, C. A., Chou, R. T., ... & Ensign, L. M. (2021). Ion-complex microcrystal formulation provides sustained deliv-ery of a multimodal kinase inhibitor from the subconjunctival space for the protection of retinal ganglion cells. Pharmaceutics, 13(5), 647].
Methodology
-Justify the choice of sunitinib dose (25 mg/kg).
Response: Sunitinib was previously used at the same dose (25 mg/kg) in different experimental studies and showed antioxidant and anti-inflammatory activity [2-4].’
2-Bilici, S.; Yazici, G.N.; Altuner, D.; Aggul, A.G, Suleyman H. Effect of Sunitinib on Liver Oxidative and Proinflammatory Damage Induced by Ischemia-Reperfusion in Rats. Transplant Proc. 2021, 53, 2140-2146.
3-Keskin, E.; Erdogan, A.; Suleyman, H.; Yazici, G.N..; Sunar, M.; Gul, M.A. Effect of sunitinib on testicular oxidative and proinflammatory damage induced by ischemia-reperfusion in rats. Rev. Int. Androl. 2022, 20 Suppl 1, S17-S23.
4-İnce, S.; Özer, M.; Göktuğ Kadıoğlu, B.; Kuzucu, M.; Gündoğdu, B.; Gürsul, C.; Süleyman Z, Suleyman, H. Biochemical and Histopathological Evaluation of Sunitinib Effect on Ovarian Injuries by Ischemia-Reperfusion in Rats. Int. J. Pharmacol. 2021, 17, 65-72.
-Were different doses tested in pilot studies? How does this dose compare with clinical equivalents?
Response: You are right about this. Sunitinib was previously used by our team in different studies and showed antioxidant and anti-inflammatory activity at the same dose (2-4). Therefore, a single-dose study was preferred. The following test was added to the experimental procedures section
‘Doses in this study were determined concerning previous studies. Drug doses administered in experimental animals differ from human doses due to differences in body surface area and metabolism. Doses used in animal studies are converted to human doses by various methods [1]. Sunitinib was previously used at the same dose (25 mg/kg) in different experimental studies and showed antioxidant and anti-inflammatory activity [2-4].’
1-Nair, A. B., and Jacob, S. (2016). A simple practice guide for dose conversion between animals and human. J. basic Clin. Pharm. 7 (2), 27–31. doi:10.4103/0976-0105.177703
2-Bilici, S.; Yazici, G.N.; Altuner, D.; Aggul, A.G, Suleyman H. Effect of Sunitinib on Liver Oxidative and Proinflammatory Damage Induced by Ischemia-Reperfusion in Rats. Transplant Proc. 2021, 53, 2140-2146.
3-Keskin, E.; Erdogan, A.; Suleyman, H.; Yazici, G.N..; Sunar, M.; Gul, M.A. Effect of sunitinib on testicular oxidative and proinflammatory damage induced by ischemia-reperfusion in rats. Rev. Int. Androl. 2022, 20 Suppl 1, S17-S23.
4-İnce, S.; Özer, M.; Göktuğ Kadıoğlu, B.; Kuzucu, M.; Gündoğdu, B.; Gürsul, C.; Süleyman Z, Suleyman, H. Biochemical and Histopathological Evaluation of Sunitinib Effect on Ovarian Injuries by Ischemia-Reperfusion in Rats. Int. J. Pharmacol. 2021, 17, 65-72.
-The reperfusion period of 6 hours is relatively short for assessing long-term neuroprotection. Would longer observation periods provide more translationally relevant insights?
Response: You are right that long-term effects after reperfusion could also be evaluated. Thank you for this reminder. We will definitely take this into account in our future studies.
Emir, I., Suleyman, Z., & Suleyman, H. (2024). Effect of thiamine pyrophosphate on oxidative damage in the brain and heart of rats with experimentally induced occlusion of the common carotid artery. Investigación Clínica, 65(2), 220-229].
Musayeva, A., Unkrig, J. C., Zhutdieva, M. B., Manicam, C., Ruan, Y., Laspas, P., Chronopoulos, P., Göbel, M. L., Pfeiffer, N., Brochhausen, C., Daiber, A., Oelze, M., Li, H., Xia, N., & Gericke, A. (2021). Betulinic Acid Protects from Ischemia-Reperfusion Injury in the Mouse Retina. Cells, 10(9), 2440. https://doi.org/10.3390/cells10092440
-The study lacks functional assessments such as visual evoked potentials or behavioral tests that could have strengthened the impact of findings beyond biochemical and histopathological analysis.
Response: Deficiencies have been added to the limitations section. Thank you for your reminder. We will take this into account in our future work.
Results Interpretation
-The biochemical findings are clear, but the statistical power of comparisons should be explicitly stated. Were power calculations performed for the sample size?
Response: The number of animals in the study was determined according to previous studies. Since the ethics committees requested the use of the minimum number of animals in accordance with the 3R rule for animal welfare, the number of animals per group was determined as 6.
Cicek, I.; Somuncu, A.M.; Altuner, D.; Suleyman, B.; Mammadov, R.; Bulut, S.; Coban, T.A.; Bal Tastan, T.; Suleyman, H. Lacidipine, thiamine pyrophosphate and their combination on the ocular ischemic syndrome induced by bilateral common carotid artery ligation. Int. J. Ophthalmol. 2024, 17, 815-821
-The study demonstrates increased contralateral optic nerve damage, which is a crucial finding. The authors should expand on the mechanism of mirror-image injury in I/R models, possibly citing recent work on cross-neuronal inflammatory signaling.
Response: The discussion of the mirror-image injury has been expanded. The following text has been added to the discussion.
‘It has been reported that experimental perfusion reduction in one eye leads to decreased bilateral retinal function in healthy subjects …………..It is an accepted view that unilateral lesions of the central nervous system or perfusion problems trigger an inflammatory response in the contralateral area. Contralateral retinas in models of unilateral retinal damage have been reported to show neuronal degeneration and glial activation [17]. Excessive glial activation threatens cell viability by creating an unstable proinflammatory environment that impairs neuronal survival.’
-The histopathological findings should be correlated better with biochemical markers. For instance, how does astrocyte proliferation relate to oxidative stress markers like MDA?
Response: Additions were made to the discussion section for the relationship between histopathological changes and biochemical changes.
Discussion
-While the discussion is well-structured, it would benefit from a comparative analysis with other neuroprotective agents tested in optic nerve ischemia (e.g., anti-inflammatory drugs, antioxidants, or VEGF inhibitors).
Response: Additions were made to the discussion section.
The limitations section should address the need for dose-dependent studies, long-term evaluation, and functional outcome measures to assess the real-world applicability of sunitinib.
Response: Deficiencies have been added to the limitations section. Thank you for your reminder. We will take this into account in our future work.
Minor Comments:
-Line 22-23: The study objective should be more precise, emphasizing sunitinib’s specific mechanisms of action.Satır 22-23:
Response: The purpose statement has been changed to the following.
‘in this study, it was investigated the effect of sunitinib, whose antioxidant and anti-inflammatory properties have been previously reported and shown to be protective in I/R injury, in preventing bilateral optic nerve ischemia-reperfusion (I/R) injuries after unilateral common carotid artery ligation in rats.’
-Line 81-82: Manufacturer details of sevoflurane should be double-checked for accuracy.Satır 81-82:
Response: Thank you for your attention. The error has been corrected.
-Table 1: Ensure consistent formatting in the presentation of statistical significance markers (p-values).
Response: Table 1 p-values have been adjusted.
-References: Some references are outdated (e.g., studies before 2010). Consider incorporating recent publications (2020-2024) on optic nerve I/R injury and pharmacological interventions.Referanslar:
Response: References updated.
Reviewer 3
Major Comments
-I suggest to include additional background on the broader clinical relevance of optic nerve ischemia , example: its relation to glaucoma or stroke-induced optic neuropathy).
Response: The following text has been added to the introduction based on your suggestion. Clinically common causes such as glaucoma and ischaemic stroke may also lead to ischemia-induced optic nerve damage. Increased intraocular pressure leads to tissue ischemia, oxidative stress, inflammation, and ultimately glaucomatous neuropathy [Coviltir, V., Burcel, M. G., Baltă, G., & Marinescu, M. C. (2024). Interplay Between Ocular Ischemia and Glaucoma: An Update. International Journal of Molecular Sciences, 25(22), 12400]. Reduced blood flow during ischaemic stroke interferes with normal mitochondrial functioning and various triggered events, including reduced energy and oxidative stress, can lead to damage of retinal ganglion cells and the optic nerve [Kingsbury, C., Heyck, M., Bonsack, B., Lee, J. Y., & Borlongan, C. V. (2020). Stroke gets in your eyes: stroke-induced retinal ischemia and the potential of stem cell therapy. Neural Regeneration Research, 15(6), 1014-1018.].
-Discuss the recent advances in the use of multikinase inhibitors like sunitinib for neuroprotection in ophthalmology.
Response: Additions were made to the discussion section.
-It is important to include referencing studies on sunitinib’s effects on retinal or central nervous system ischemia models for better contextualization.
Response: Studies on the neuroprotective effect of sunitibib against ischemia were added to the introduction section. The following text has been added to the introduction section.
However, the neuroprotective effect of sunitinib has been previously investigated and shown to promote retinal ganglion cell survival in an experimental anterior ischemic optic neuropathy model [Sharma, S.M.; Wu-chen, W.Y.; Yang, Z.; Vilson, F.L.; Guo, Y.; Miller, N.R.; Don-ald, Z.J.; Bernstein, S.L. (2011). Sunitinib Malate Preserves Retinal Ganglion Cells in Rodent NAION. Invest. Ophthalmol. Vis. Sci. 2011, 52, 6614-6614]. A single subconjunctival injection of sunitinib-pamoate complex microcrystals was shown to provide neuroprotection in a rat optic nerve crush model [Hsueh, H. T., Kim, Y. C., Pitha, I., Shin, M. D., Berlinicke, C. A., Chou, R. T., ... & Ensign, L. M. (2021). Ion-complex microcrystal formulation provides sustained deliv-ery of a multimodal kinase inhibitor from the subconjunctival space for the protection of retinal ganglion cells. Pharmaceutics, 13(5), 647].
Methodology
-Justify the choice of sunitinib dose (25 mg/kg).
Response: Sunitinib was previously used at the same dose (25 mg/kg) in different experimental studies and showed antioxidant and anti-inflammatory activity [2-4].’
2-Bilici, S.; Yazici, G.N.; Altuner, D.; Aggul, A.G, Suleyman H. Effect of Sunitinib on Liver Oxidative and Proinflammatory Damage Induced by Ischemia-Reperfusion in Rats. Transplant Proc. 2021, 53, 2140-2146.
3-Keskin, E.; Erdogan, A.; Suleyman, H.; Yazici, G.N..; Sunar, M.; Gul, M.A. Effect of sunitinib on testicular oxidative and proinflammatory damage induced by ischemia-reperfusion in rats. Rev. Int. Androl. 2022, 20 Suppl 1, S17-S23.
4-İnce, S.; Özer, M.; Göktuğ Kadıoğlu, B.; Kuzucu, M.; Gündoğdu, B.; Gürsul, C.; Süleyman Z, Suleyman, H. Biochemical and Histopathological Evaluation of Sunitinib Effect on Ovarian Injuries by Ischemia-Reperfusion in Rats. Int. J. Pharmacol. 2021, 17, 65-72.
-Were different doses tested in pilot studies? How does this dose compare with clinical equivalents?
Response: You are right about this. Sunitinib was previously used by our team in different studies and showed antioxidant and anti-inflammatory activity at the same dose (2-4). Therefore, a single-dose study was preferred. The following test was added to the experimental procedures section
‘Doses in this study were determined concerning previous studies. Drug doses administered in experimental animals differ from human doses due to differences in body surface area and metabolism. Doses used in animal studies are converted to human doses by various methods [1]. Sunitinib was previously used at the same dose (25 mg/kg) in different experimental studies and showed antioxidant and anti-inflammatory activity [2-4].’
1-Nair, A. B., and Jacob, S. (2016). A simple practice guide for dose conversion between animals and human. J. basic Clin. Pharm. 7 (2), 27–31. doi:10.4103/0976-0105.177703
2-Bilici, S.; Yazici, G.N.; Altuner, D.; Aggul, A.G, Suleyman H. Effect of Sunitinib on Liver Oxidative and Proinflammatory Damage Induced by Ischemia-Reperfusion in Rats. Transplant Proc. 2021, 53, 2140-2146.
3-Keskin, E.; Erdogan, A.; Suleyman, H.; Yazici, G.N..; Sunar, M.; Gul, M.A. Effect of sunitinib on testicular oxidative and proinflammatory damage induced by ischemia-reperfusion in rats. Rev. Int. Androl. 2022, 20 Suppl 1, S17-S23.
4-İnce, S.; Özer, M.; Göktuğ Kadıoğlu, B.; Kuzucu, M.; Gündoğdu, B.; Gürsul, C.; Süleyman Z, Suleyman, H. Biochemical and Histopathological Evaluation of Sunitinib Effect on Ovarian Injuries by Ischemia-Reperfusion in Rats. Int. J. Pharmacol. 2021, 17, 65-72.
-The reperfusion period of 6 hours is relatively short for assessing long-term neuroprotection. Would longer observation periods provide more translationally relevant insights?
Response: You are right that long-term effects after reperfusion could also be evaluated. Thank you for this reminder. We will definitely take this into account in our future studies.
Emir, I., Suleyman, Z., & Suleyman, H. (2024). Effect of thiamine pyrophosphate on oxidative damage in the brain and heart of rats with experimentally induced occlusion of the common carotid artery. Investigación Clínica, 65(2), 220-229].
Musayeva, A., Unkrig, J. C., Zhutdieva, M. B., Manicam, C., Ruan, Y., Laspas, P., Chronopoulos, P., Göbel, M. L., Pfeiffer, N., Brochhausen, C., Daiber, A., Oelze, M., Li, H., Xia, N., & Gericke, A. (2021). Betulinic Acid Protects from Ischemia-Reperfusion Injury in the Mouse Retina. Cells, 10(9), 2440. https://doi.org/10.3390/cells10092440
-The study lacks functional assessments such as visual evoked potentials or behavioral tests that could have strengthened the impact of findings beyond biochemical and histopathological analysis.
Response: Deficiencies have been added to the limitations section. Thank you for your reminder. We will take this into account in our future work.
Results Interpretation
-The biochemical findings are clear, but the statistical power of comparisons should be explicitly stated. Were power calculations performed for the sample size?
Response: The number of animals in the study was determined according to previous studies. Since the ethics committees requested the use of the minimum number of animals in accordance with the 3R rule for animal welfare, the number of animals per group was determined as 6.
Cicek, I.; Somuncu, A.M.; Altuner, D.; Suleyman, B.; Mammadov, R.; Bulut, S.; Coban, T.A.; Bal Tastan, T.; Suleyman, H. Lacidipine, thiamine pyrophosphate and their combination on the ocular ischemic syndrome induced by bilateral common carotid artery ligation. Int. J. Ophthalmol. 2024, 17, 815-821
-The study demonstrates increased contralateral optic nerve damage, which is a crucial finding. The authors should expand on the mechanism of mirror-image injury in I/R models, possibly citing recent work on cross-neuronal inflammatory signaling.
Response: The discussion of the mirror-image injury has been expanded. The following text has been added to the discussion.
‘It has been reported that experimental perfusion reduction in one eye leads to decreased bilateral retinal function in healthy subjects …………..It is an accepted view that unilateral lesions of the central nervous system or perfusion problems trigger an inflammatory response in the contralateral area. Contralateral retinas in models of unilateral retinal damage have been reported to show neuronal degeneration and glial activation [17]. Excessive glial activation threatens cell viability by creating an unstable proinflammatory environment that impairs neuronal survival.’
-The histopathological findings should be correlated better with biochemical markers. For instance, how does astrocyte proliferation relate to oxidative stress markers like MDA?
Response: Additions were made to the discussion section for the relationship between histopathological changes and biochemical changes.
Discussion
-While the discussion is well-structured, it would benefit from a comparative analysis with other neuroprotective agents tested in optic nerve ischemia (e.g., anti-inflammatory drugs, antioxidants, or VEGF inhibitors).
Response: Additions were made to the discussion section.
The limitations section should address the need for dose-dependent studies, long-term evaluation, and functional outcome measures to assess the real-world applicability of sunitinib.
Response: Deficiencies have been added to the limitations section. Thank you for your reminder. We will take this into account in our future work.
Minor Comments:
-Line 22-23: The study objective should be more precise, emphasizing sunitinib’s specific mechanisms of action.Satır 22-23:
Response: The purpose statement has been changed to the following.
‘in this study, it was investigated the effect of sunitinib, whose antioxidant and anti-inflammatory properties have been previously reported and shown to be protective in I/R injury, in preventing bilateral optic nerve ischemia-reperfusion (I/R) injuries after unilateral common carotid artery ligation in rats.’
-Line 81-82: Manufacturer details of sevoflurane should be double-checked for accuracy.Satır 81-82:
Response: Thank you for your attention. The error has been corrected.
-Table 1: Ensure consistent formatting in the presentation of statistical significance markers (p-values).
Response: Table 1 p-values have been adjusted.
-References: Some references are outdated (e.g., studies before 2010). Consider incorporating recent publications (2020-2024) on optic nerve I/R injury and pharmacological interventions.Referanslar:
Response: References updated.
-Oftalmolojide nöroproteksiyon için sunitinib gibi multikinaz inhibitörlerinin kullanımındaki son gelişmeler.

Round 2
Reviewer 1 Report
Comments and Suggestions for Authors
The author answered all my questions and made careful revisions in the revised manuscript. The quality of the revised manuscript has been improved. Due to some issues that the author agrees with, it is suggested to add a limitation section in the discussion.
Comments on the Quality of English LanguageI haven't found any major issues with the English language, which doesn't hinder my reading.
Author Response
Reviewer 1
Comments 1. The author answered all my questions and made careful revisions in the revised manuscript. The quality of the revised manuscript has been improved. Due to some issues that the author agrees with, it is suggested to add a limitation section in the discussion.
Response: The discussion section was reviewed and new limitations were added.
‘Limitations
Dose-dependent studies, long-term evaluation, and functional outcome measures are important to assess the feasibility of sunitinib. Furthermore, the protective effect of sunitinib can be evaluated by specific cellular and molecular mechanisms. Limitations of the study include the lack of functional assessments such as visual evoked potentials or behavioral tests.’
Doza bağlı çalışmalar, uzun süreli değerlendirme ve fonksiyonel sonuç ölçümleri, sunitinibin fizibilitesini değerlendirmek için önemlidir.
